# Prediction Under Uncertainty with Error-Encoding Networks

## Abstract

In this work we introduce a new framework for performing temporal predictions in the presence of uncertainty. It is based on a simple idea of disentangling components of the future state which are predictable from those which are inherently unpredictable, and encoding the unpredictable components into a low-dimensional latent variable which is fed into a forward model. Our method uses a supervised training objective which is fast and easy to train. We evaluate it in the context of video prediction on multiple datasets and show that it is able to consistently generate diverse predictions without the need for alternating minimization over a latent space or adversarial training.

## 1 Introduction

Learning forward models in time series is a central task in artificial intelligence, with applications in unsupervised learning, planning and compression. A major challenge in this task is how to handle the multi-modal nature of many time series. When there are multiple valid ways in which a time series can evolve, training a model using classical $\ell_1$ or $\ell_2$ losses produces predictions which are the average or median of the different outcomes across each dimension, which is itself often not a valid prediction.

In recent years, Generative Adversarial Networks (Goodfellow et al., 2014) have been introduced, a general framework where the prediction problem is formulated as a minimax game between the predictor function and a trainable discriminator network representing the loss. By using a trainable loss function, it is in theory possible to handle multiple output modes since a generator which covers each of the output modes will fool the discriminator leading to convergence. However, a generator which covers a single mode can also fool the discriminator and converge, and this behavior of mode collapse has been widely observed in practice. Some workarounds have been introduced to resolve or partially reduce mode-collapsing, such as minibatch discrimination, adding parameter noise (Salimans et al., 2016), backpropagating through the unrolled discriminator (Metz et al., 2016) and using multiple GANs to cover different modes (Tolstikhin et al., 2017). However, many of these techniques can bring additional challenges such as added complexity of implementation and increased computational cost. The mode collapsing problem becomes even more pronounced in the conditional generation setting when the output is highly dependent on the context, such as video prediction (Mathieu et al., 2015; Isola et al., 2016).

In this work, we introduce a novel architecture that allows for robust multimodal conditional predictions in time series data. It is based on a simple intuition of separating the future state into a deterministic component, which can be predicted from the current state, and a stochastic (or difficult to predict) component which accounts for the uncertainty regarding the future mode. By training a deterministic network, we can obtain this factorization in the form of the network's prediction together with the prediction error with respect to the true state. This error can be encoded as a low-dimensional latent variable which is fed into a second network which is trained to accurately correct the determinisic prediction by incorporating this additional information. We call this model the Error Encoding Network (EEN). In a nutshell, this framework contains three function mappings at each timestep: (i) a mapping from the current state to the future state, which separates the future state into deterministic and non-deterministic components; (ii) a mapping from the non-deterministic component of the future state to a low-dimensional latent vector; (iii) a mapping from the current state to the future state conditioned on the latent vector, which encodes the mode information of the future

state. While the training procedure involves all these mappings, the inference phase involves only (iii).

Both networks are trained end-to-end using a supervised learning objective and latent variables are computed using a learned parametric function, leading to easy and fast training. We apply this method to video datasets from games, robotic manipulation and simulated driving, and show that the method is able to consistently produce multimodal predictions of future video frames for all of them. Although we focus on video in this work, the method itself is general and can in principle be applied to any continuous-valued time series.

## 2 MODEL

Many natural processes carry some degree of uncertainty. This uncertainty may be due to an inherently stochastic process, a deterministic process which is partially observed, or it may be due to the complexity of the process being greater than the capacity of the forward model. One natural way of dealing with uncertainty is through latent variables, which can be made to account for aspects of the target that are not explainable from the observed input.

Assume we have a set of continuous vector-valued input-target pairs $(x_i, y_i)$, where the targets depend on both the inputs and some inherently unpredictable factors. For example, the inputs could be a set of consecutive video frames and the target could be the following frame. Classical latent variable models such as $k$-means or mixtures of Gaussians are trained by alternately minimizing the loss with respect to the latent variables and model parameters; in the probabilistic case this is the Expectation-Maximization algorithm (Dempster et al., 1977). In the case of a neural network model $f_\theta(x_i, z)$, continuous latent variables can be optimized using gradient descent and the model can be trained with the following procedure:

---
**Algorithm 1** Train latent variable model with alternating minimization
---
**Require:** Learning rates $\alpha, \beta$, number of iterations $K$.
 1: **repeat**
 2:     Sample $(x_i, y_i)$ from the dataset
 3:     initialize $z \sim \mathcal{N}(0, 1)$
 4:     $i \leftarrow 1$
 5:     **while** $i \leq K$ **do**
 6:         $z \leftarrow z - \alpha \nabla_z \mathcal{L}(y_i, f_\theta(x_i, z))$
 7:         $i \leftarrow i + 1$
 8:     $\theta \leftarrow \theta - \beta \nabla_\theta \mathcal{L}(y_i, f_\theta(x_i, z))$
 9: **until** converged
---

Our approach is based on two observations. First, the latent variable $z$ should represent what is not explainable using the input $x_i$. Ideally, the model should make use of the input $x_i$ and only use $z$ to account for what is not predictable from it. Second, if we are using gradient descent to optimize the latent variables, $z$ will be a continuous function of $x_i$ and $y_i$, although a possibly highly nonlinear one. We train two functions to predict the targets: a deterministic model $g(x)$ and a latent variable model $f(x, z)$. We first train the deterministic model $g$ to minimize the following loss over the training set:

$$\mathcal{L}_g = \sum_i \|y_i - g(x_i)\| \tag{1}$$

Here the norm can denote $\ell_1$, $\ell_2$ or any other loss which is a function of the difference between the target and the prediction. Given sufficient data and capacity, $g$ will learn to extract all the information possible about each $y_i$ from the corresponding $x_i$, and what is inherently unpredictable will be contained within the residual error, $y_i - g(x_i)$. This can be passed through a learned parametric function $\phi$ to produce a low-dimensional latent variable $z = \phi(y_i - g(x_i))$ which encodes the identity of the mode to which the future state belongs. This can then be used as input to $f$ to more accurately predict $y_i$ conditioned on knowledge of the proper mode.

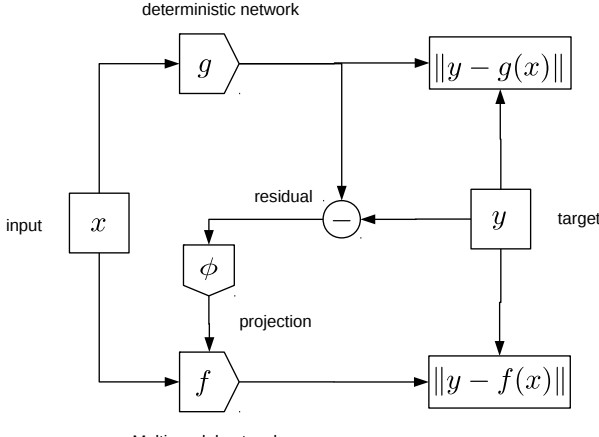

Figure 1: Model Architecure.

Once $g$ is trained, we then minimize the following loss over the training data:

$$\mathcal{L}_f = \sum_i \|y_i - f(x_i, \phi(y_i - g(x_i)))\| \tag{2}$$

The fact that $z$ is a function of the prediction error of $g$ reflects the intuition that it should only account for what is not explainable by the input, while still being a smooth function of $x$ and $y$. Note that $z$ is typically of much lower dimension than $y_i - g(x_i)$, which prevents the network from learning a trivial solution where $f$ would simply invert $\phi$ and cancel the error from the prediction. This forces the network to map the errors to general representations which can be reused across different samples and correspond to different modes of the conditional distribution.

To perform inference after the network is trained, we first extract and save the $z_i = \phi(y_i - g(x_i))$ from each sample in the training set. Given some new input $x'$, we can then generate different predictions by computing $f(x', z')$, for different $z'$ sampled from the set $\{z_i\}$. This can be seen as a non-parametric estimation of the distribution over $z$, which differs from other approaches such as VAEs (Kingma & Welling) where $z$ is pushed towards a predefined distribution (for example, an isotropic Gaussian) through an additional term in the loss function.

The model architecture is shown in Figure 1. To speed up training, we choose $f$ and $g$ to have similar architecture and initialize $f$ with the parameters of $g$. For example, we can have $g(x) = g_2(g_1(x))$ and $f(x, z) = f_2(f_1(x) + Wz)$ and initialize $f_1, f_2$ with the weights of $g_1, g_2$ respectively. Thus, if $\phi$ and $W$ are initialized such that $Wz$ has zero mean and small variance, the network is already able to extract all the signal from the input $x$ and can only improve further by adapting its weights to make use of the mode information contained in $z$.

## 3 RELATED WORK

In recent years a number of works have explored video prediction. These typically train models to predict future frames with the goal of learning representations which disentangle factors of variation and can be used for unsupervised learning (Srivastava et al., 2015; Villegas et al., 2017; Denton & Birodkar, 2017), or learn action-conditional forward models which can be used for planning (Oh et al., 2015; Finn et al., 2016; Agrawal et al., 2016; Kalchbrenner et al., 2016). In the first case, the predictions are deterministic and ignore the possibly multimodal nature of the time series. In the second, it is possible to make different predictions about the future by conditioning on different actions, however this requires that the training data includes additional action labels. Our work

makes different predictions about the future by conditioning on latent variables which are extracted in an unsupervised manner from the videos themselves.

Several works have used adversarial losses in the context of video prediction. The work of (Mathieu et al., 2015) used a multiscale architecture and a combination of several different losses to predict future frames in natural videos. They found that the addition of the adversarial loss and a gradient difference loss improved the generated image quality, in particular by reducing the blur effects which are common when using $\ell_2$ loss. However, they also note that the generator learns to ignore the noise and produces similar outputs to a deterministic model trained without noise. This observation was also made by (Isola et al., 2016) when training conditional networks to perform image-to-image translation.

Other works have used models for video prediction where latent variables are inferred using alternating minimization. The model in (Vondrick et al., 2015) includes a discrete latent variable which was used to choose between several different networks for predicting hidden states of future video frames obtained using a pretrained network. This is more flexible than a purely deterministic model, however the use of a discrete latent variable still limits the possible future modes to a discrete set. The work of (Goroshin et al., 2015) also made use of latent variables to model uncertainty, which were inferred through alternating minimization. In contrast, our model infers continuous latent variables through a learned parametric function. This is related to algorithms which learn to predict the solution of an iterative optimization procedure (Gregor & LeCun, 2010).

Recent work has shown that good generative models can be learned by jointly learning representations in a latent space together with the parameters of a decoder model (Bojanowski et al., 2017). This leads to easier training than adversarial networks. This generative model is also learned by alternating minimization over the latent variables and parameters of the decoder model, however the latent variables for each sample are saved after each update and optimization resumes when the corresponding sample is drawn again from the training set. This is related to our method, with the difference that rather than saving the latent variables for each sample we compute them through a learned function of the deterministic network's prediction error.

Our work is related to predictive coding models (Rao & Ballard, 1999; Spratling, 2008; Chalasani & Principe, 2013; Lotter et al., 2016) and chunking architectures (Schmidhuber, 1992), which also pass residual errors or incorrectly predicted inputs between different parts of the network. It differs in that these models pass errors upwards to higher layers in the network at each timestep, whereas our model passes the compressed error signal from the deterministic network backwards in time to serve as input to the latent variable network at the previous timestep.

Our model is also related to Variational Autoencoders (Kingma & Welling), in that we use a latent variable $z$ at training time which is a function of the input $x$ and target $y$. It differs in the use of a pretrained deterministic model which is used to compute the residual error of which $z$ is a function and for initializing the network, as well as the non-parametric sampling procedure which does not place any prior assumptions on the $z$ distribution and removes the need for an additional term in the loss function enforcing that the $z$ distribution matches this prior.

## 4 EXPERIMENTS

We tested our method on four different video datasets from different areas such as games (Atari Breakout and Flappy Bird), robot manipulation (Agrawal et al., 2016) and simulated driving (Zhang & Cho, 2016). These have a well-defined multimodal structure, where the environment can change due to the actions of the agent or other stochastic factors and span a diverse range of visual environments. For each dataset, we trained our model to predict the next 4 frames conditioned on the previous 4 frames. Code to train our models and obtain video generations is available at `url`.

We trained several baseline models to compare performance: a deterministic model (CNN), a model with $z$ variables drawn from an isotropic Gaussian (CNN + noise), a conditional autoencoder (AE), a conditional variational autoencoder model (cVAE), and a GAN. These different models are summarized in Table 1. Note that they have the same architecture but differ in their loss function and/or sampling procedure for the $z$ variables. We used the same architecture across all tasks and for all models, namely $f_2(f_1(x) + Wz)$ where $f_1$ is a 3-layer convolutional network and $f_2$ is a 3-layer deconvolutional network, all with 64 feature maps at each layer and batch normalization. We did

| Model | $z$ (train) | $z$ (test) | Loss |
|---|---|---|---|
| CNN | $z = 0$ | $z = 0$ | $\|\tilde{y} - y\|$ |
| CNN + noise | $z \sim \mathcal{N}(0, I)$ | $z \sim \mathcal{N}(0, I)$ | $\|\tilde{y} - y\|$ |
| cVAE | $z \sim \mathcal{N}(\mu, \sigma)$ where $(\mu, \sigma) = \phi(x, y)$ | $z \sim \mathcal{N}(0, I)$ | $\|\tilde{y} - y\| + KL(\mathcal{N}(\mu, \sigma), \mathcal{N}(0, I))$ |
| GAN | $z \sim \mathcal{N}(0, I)$ | $z \sim \mathcal{N}(0, I)$ | Discriminator $D$ |
| AE | $z = \phi(x, y)$ | $z \sim \{z_i\}$ | $\|\tilde{y} - y\|$ |
| EEN | $z = \phi(y - g(x))$ | $z \sim \{z_i\}$ | $\|\tilde{y} - y\|$ |

Table 1: Inference methods and loss for the different models. All models have the same architecture, $\tilde{y} = f_2(f_1(x) + Wz)$.

not use pooling and instead used strided convolutions, similar to the DCGAN architecture (Radford et al., 2015).

The $\phi$ network represents the posterior network for the cVAE, the encoder network for the AE and the network described in Section 2 for the EEN. For all models $\phi$ had a similar structure, i.e. a 3-layer convolutional network with 64 feature maps followed by two fully-connected layers with 1000 units each which output the $z$ variables for the EEN and AE and the parameters of the $z$ distribution for the CVAE. All models except the GAN were trained using the $\ell_2$ loss for all datasets except the Robot dataset, where we found that the $\ell_1$ loss gave better-defined predictions. Although more sophisticated losses exist, such as the Gradient Difference loss (Mathieu et al., 2015), our goal here was to evaluate whether our model could capture multimodal structure such as objects moving or appearing on the screen or perspective changing in multiple different realistic ways.

The EEN, AE, CNN and CNN + noise were trained using the ADAM optimizer (Kingma & Ba, 2014) with default parameters and learning rate 0.0005 for all tasks. For the cVAE, we additionally optimized learning rates over the range $\{0.005, 0.001, 0.0005, 0.0001\}$. For these models we used 8 latent variables on all tasks except for the driving task, where we used 32. For the GAN, based on visual examination of the generated samples, we selected the learning rates of the generator and discriminator to be 0.00005 and 0.00001 respectively. We used 64 latent variables in the GAN experiments. The other settings with GAN experiments are designed similarly to DCGAN's, such as using Leaky ReLU in the discriminator, Batch Normalization and etc.

## 4.1 DATASETS

We now describe the video datasets we used.

**Atari Games** We used a pretrained A2C agent (Mnih et al., 2016) [1] to generate episodes of gameplay for the Atari games Breakout (Bellemare et al., 2012) using a standard video preprocessing pipeline, i.e. downsampling video frames to $84 \times 84$ pixels and converting to grayscale. We then trained our forward model using 4 consecutive frames as input to predict the following 4 frames.

**Flappy Bird** We used the OpenAI Gym environment Flappy Bird [2] and had a human player play approximately 50 episodes of gameplay. In this environment, the player controls a moving bird which must navigate between obstacles appearing at different heights. We trained the model to predict the next 4 frames using the previous 4 frames as input, all of which were rescaled to $128 \times 72$ pixel color images.

**Robot Manipulation** We used the dataset of (Agrawal et al., 2016), which consists of $240 \times 240$ pixel color images of objects on a table before and after manipulation by a robot. The robot pokes the object at a random location with random angle and duration causing it to move, hence the manipulation does not depend of the environment except for the location of the object. Our model was trained to take a single image as input and predict the following image.

**Simulated Driving** We used the dataset from (Zhang & Cho, 2016), which consists of color videos from the front of a car taken within the TORCS simulated driving environment. This car is driven by an agent whose policy is to follow the road and pass or avoid other cars while staying within the

---

[1] https://github.com/ikostrikov/pytorch-a2c-ppo-acktr
[2] https://gym.openai.com/envs/FlappyBird-v0/

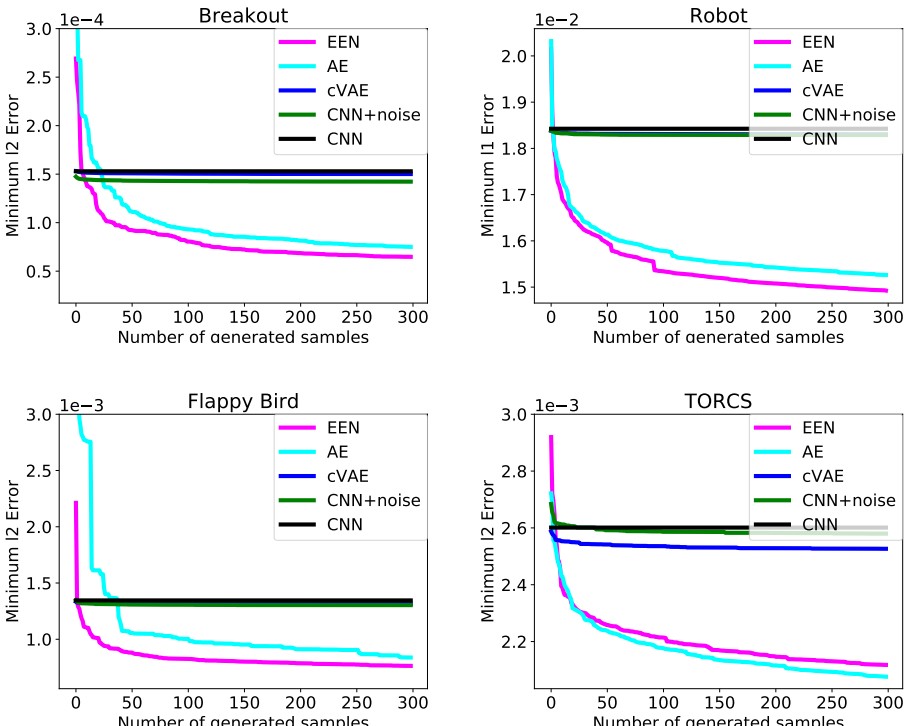

Figure 2: Best loss for different models over varying numbers of different samples.

speed limit. Here we again trained the model to predict 4 frames using the 4 previous frames as input. Each image was rescaled to $160 \times 72$ pixels as in the original work.

## 4.2 RESULTS

Our experiments were designed to test whether our method can generate multiple realistic predictions given the start of a video sequence. We first report quantitative results, using the approach adopted in (Walker et al., 2016). As noted by the authors, quantitatively evaluating multimodal predictions is challenging, since the ground truth sample is drawn from one of several possible modes and the model may generate a sample from a different mode. In this case, simply comparing the generated sample to the ground truth sample may give high loss even if the generated sample is of high quality. We therefore fix a number of samples each model is allowed to generate and report the best score across different generated samples: $\min_{k} \mathcal{L}(y, f(x, z_k))$. If the multimodal model is able to use its latent variables to generate predictions which cover several modes, generating more samples will improve the score since it increases the chance that a generated sample will be from the same mode as the test sample. If however the model ignores latent variables or does not capture the mode that the test sample is drawn from, generating more samples will not improve the loss. Note that if $\mathcal{L}$ is a valid metric in the mathematical sense (such as the $\ell_1$ or $\ell_2$ distance), this is a finite-sample approximation to the Earth Mover or Wasserstein-1 distance between the true and generated distributions on the metric space induced by $\mathcal{L}$. Although this metric can reflect how many real modes the generative model is able to cover, and thus detect mode collapse, it does not necessarily detect when the model is generating samples from a mode that is not present in the testing data. [3]

---

[3] For example, say $p(y|x)$ has modes $m_1, m_2$ with equal mass and the model generates samples in $m_1, m_2$ and a third mode $m_3$ with equal probability. This model would still get a fairly good score by our metric (although the curves would likely improve a bit more slowly with more samples). However if the model only generates samples in $m_1$ or $m_2$, it would be penalized more.

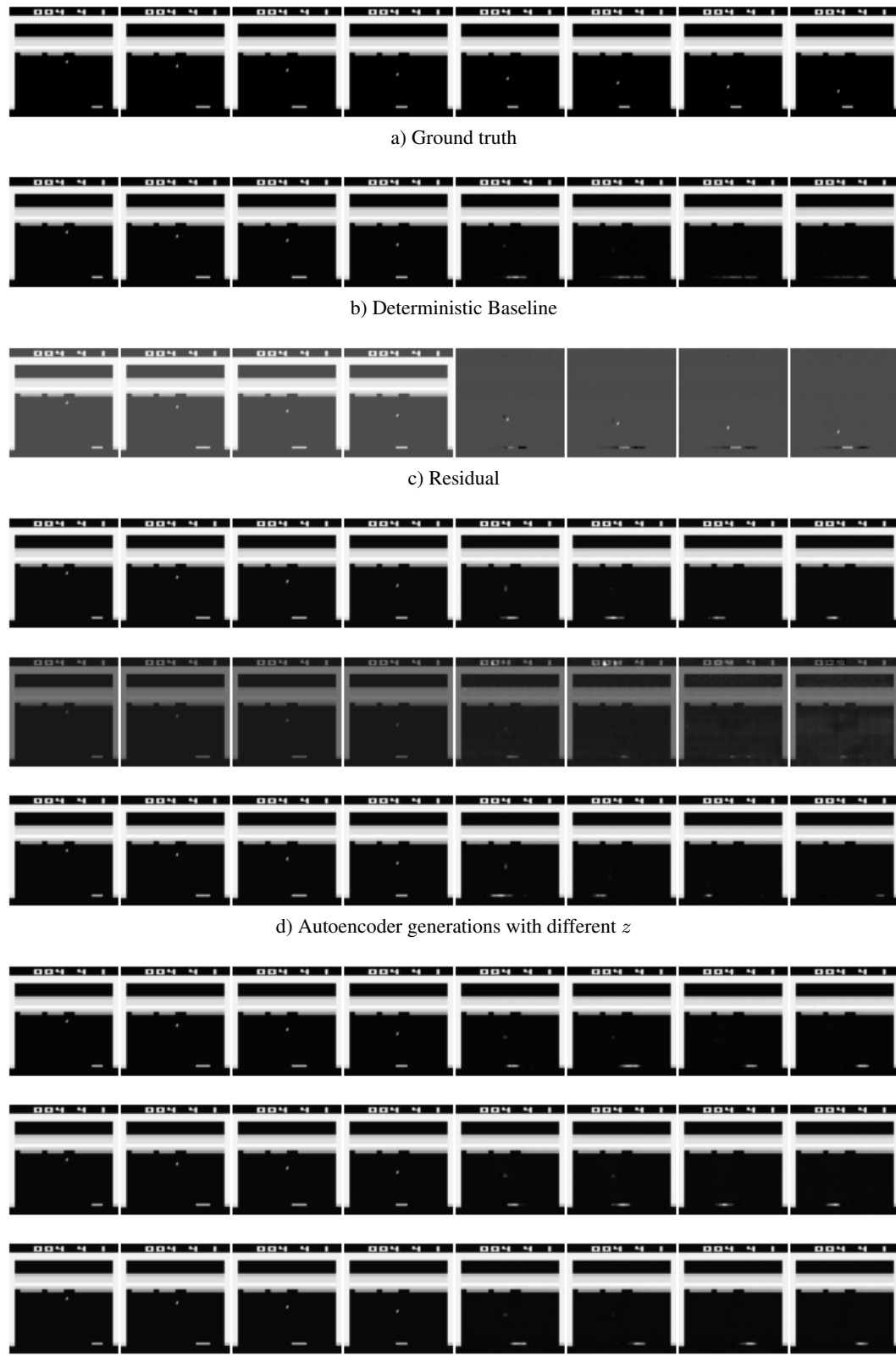

Figure 3: Generations on Breakout. Left 4 frames are given, right 4 frames are generated. Note that the paddle changes location for the different generations. Best viewed with zoom.

Figure 2 shows the best loss for different numbers of generated samples for the various models. We did not include GAN results in these plots as we found that their $\ell_1/\ell_2$ loss differed significantly from that of the other models, which skewed the plots. This is reasonable since the GANs are optimizing a different loss than the other models. Plots including the GAN performance can be found in the Appendix. The EEN and the AE see their performance increase with higher numbers of generated samples whereas the CNN + noise and cVAE have very similar performance to the baseline CNN model, indicating that they have learned to ignore their latent variables. This behavior has been observed in other works on conditional generation and is referred to as "posterior collapse". When the conditioning variable $x$ contains a lot of signal, the model can achieve large improvements in the prediction loss by learning a deterministic function of $x$, while letting the $z$ distribution go to $\mathcal{N}(0, 1)$ to lower the KL term in the loss, and having the rest of the network ignore $z$. Various strategies have been proposed such as adding additional features or losses (Zhao et al., 2017), annealing the magnitude of the KL term during training (Bowman et al., 2015; Fraccaro et al., 2016) or using discrete latent codes (van den Oord et al., 2017). In contrast, the EEN and AE are able to generate diverse predictions without any modifications or additional hyperparameters to tune, due to the non-parametric sampling procedure which does not place any assumptions on the latent distribution and does not require an additional KL term in the loss. The fact that performance increases with more generated samples indicates that the generations of the EEN and AE are diverse enough to cover at least some of the modes of the test set. Given the same number of samples, the AE performance is consistently lower than the EEN on all datasets except for TORCS, where it is comparable or slightly better depending on how many samples are generated.

We next report qualitative results in the form of visualizations. In addition to the figures in this paper, we provide a link to videos which facilitate viewing [4]. An example of generated frames in Atari Breakout is shown in Figure 3. For the baseline model, the image of the paddle gets increasingly diffuse over time which reflects the model's uncertainty as to its future location while the static background remains well defined. The residual, which is the difference between the ground truth and the baseline prediction, only depicts the movement of the ball and the paddle which the deterministic model is unable to predict. This is encoded into the latent variables $z$ through the learned function $\phi$ which takes the residual as input. By sampling different $z$ vectors from the training set, we obtain three different generations for the same conditioning frames. For these we see a well-defined paddle executing different movement sequences starting from its initial location. The autoencoder model is also able to generate diverse predictions, however it often generates predictions which contain artifacts as seen in the second set of frames.

Figure 4 shows generated frames on Flappy Bird. Flappy Bird is a simple game which is deterministic except for two sources of stochasticity: the actions of the player and the height of new pipes appearing on the screen. In this example, we see that by changing the latent variable the EEN generates two sequences with pipes entering at different heights. The autoencoder generates two sequences with pipes of different color, which is not a realistic prediction. This suggests that the $z$ variable encodes information which could be predicted from the input. In contrast, the EEN limits the information content of the $z$ variable by subtracting information which could be predicted from the input.

We next evaluated our method on the Robot dataset. For this dataset the robot pokes the object with random direction and force which cannot be predicted from the current state. The prediction of the baseline model blurs the object but does not change its location or angle. In contrast, the EEN is able to produce a diverse set of predictions where the object is moved to different adjacent locations, as shown in Figure 5. We found that the autoencoder predictions were qualitatively similar to the EEN predictions despite the fact that they scored less according to the metric in Figure 2, and therefore omit them due to space constraints. Generations from both models can be viewed at the video link.

The last dataset we evaluated our method on was the TORCS driving simulator. Here we found that generating frames with different $z$ samples changed the location of stripes on the road, and also produced translations of the frame as would happen when turning the steering wheel. We did not notice significant qualitative differences in the generations of the EEN and the AE, which is consistent with the quantitative metric in Figure 2. These effects are best viewed though the video link.

---

[4] https://sites.google.com/view/errorencodingnetworks

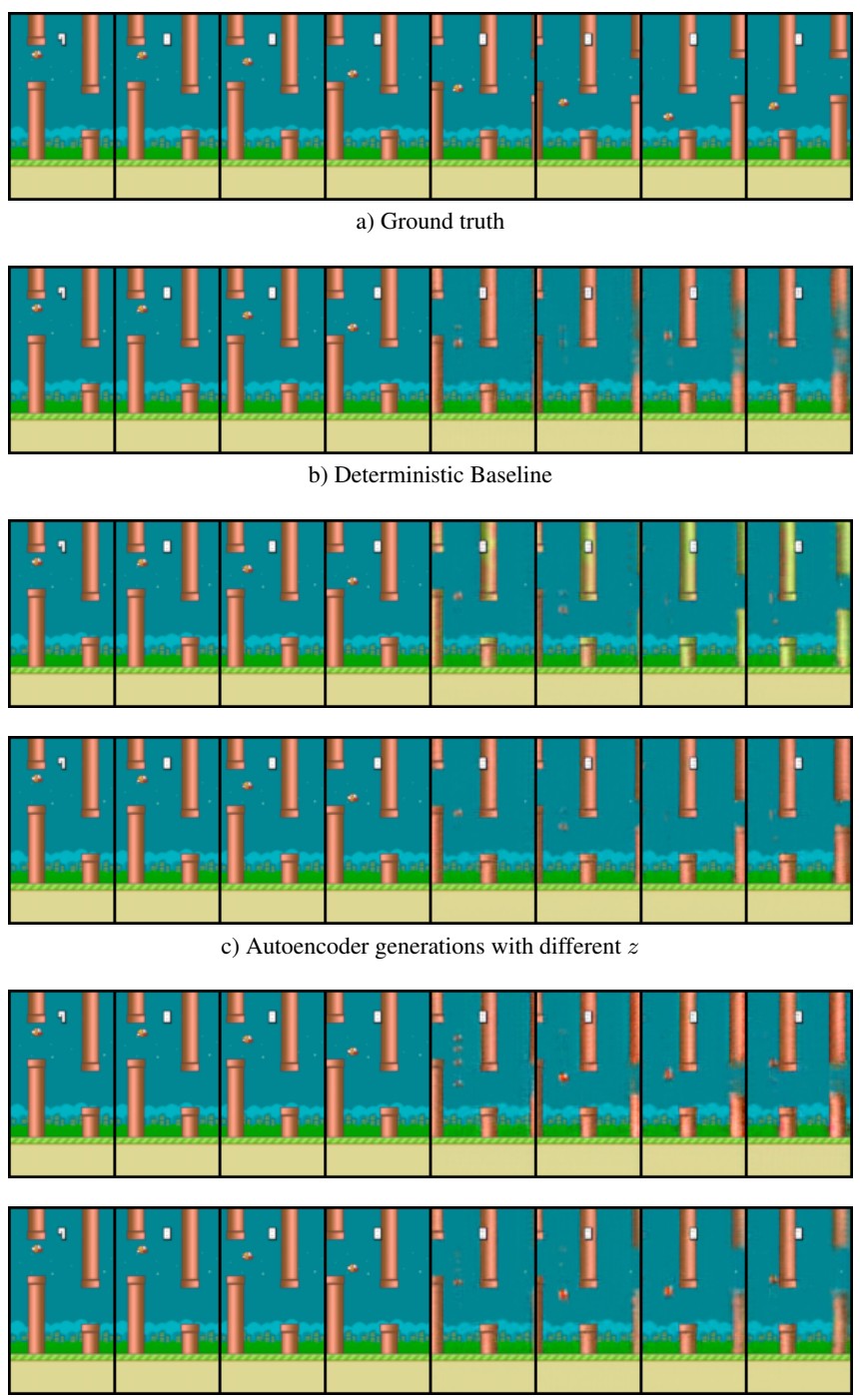

Figure 4: Generations on Flappy Bird. Left 4 frames are given, right 4 frames are generated. Note that the pipe on the right changes height for different generations. Best viewed with zoom.

## 5 CONCLUSION

In this work, we have introduced a new framework for performing temporal prediction in the presence of uncertainty by disentangling predictable and non-predictable components of the future state. It is fast, simple to implement and easy to train without the need for an adverserial network or al-

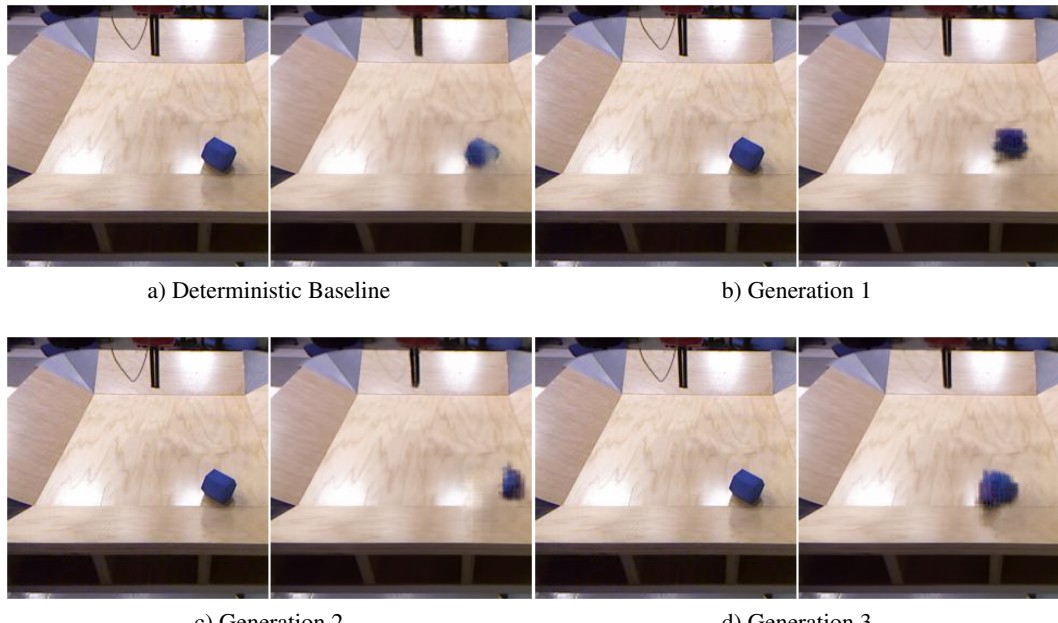

a) Deterministic Baseline b) Generation 1

c) Generation 2 d) Generation 3

Figure 5: EEN generations on Robot Task. Left frame is given, right frame is generated.

ternating minimization, and does not require additional tuning to prevent mode collapse. We have provided one instantiation in the context of video prediction using convolutional networks, but it is in principle applicable to different data types and architectures. There are several directions for future work. Here, we have adopted a simple strategy of sampling uniformly from the $z$ distribution without considering their possible dependence on the state $x$, and there are likely better methods. In addition, one advantage of our model is that it can extract latent variables from unseen data very quickly, since it simply requires a forward pass through a network. If latent variables encode information about actions in a manner that is easy to disentangle, this could be used to extract actions from large unlabeled datasets and perform imitation learning. Another interesting application would be using this model for planning and having it unroll different possible futures.

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

## 6 APPENDIX

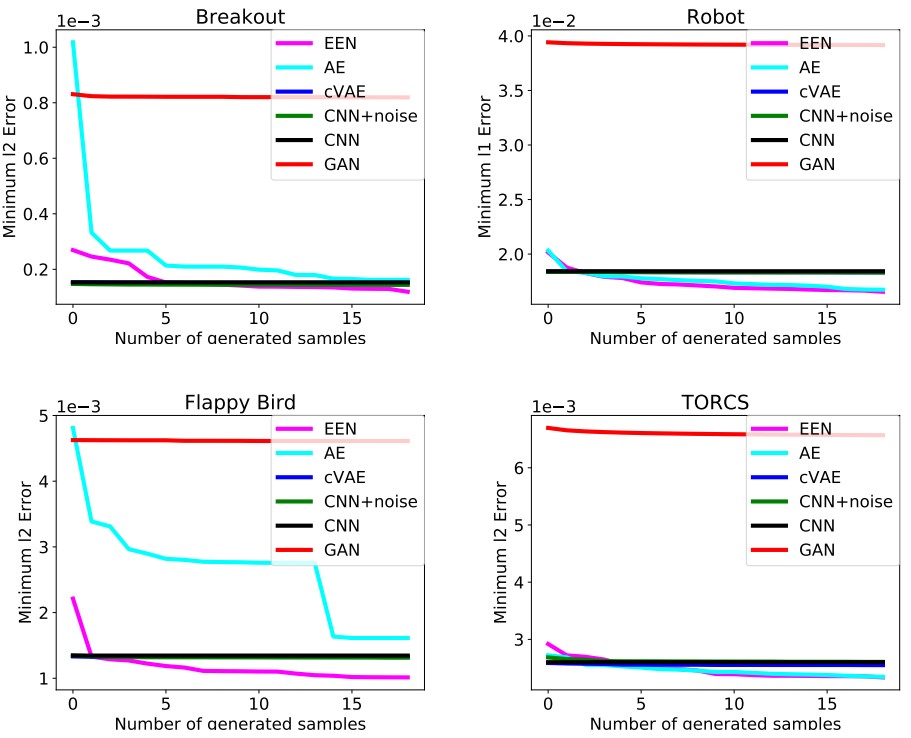

Figure 6: Best loss for different models (including GANs) over varying numbers of different samples.

