# OpenReview forum: "Prediction Under Uncertainty with Error Encoding Networks"
_ICLR.cc/2018/Conference — Reject_

### Official Review · AnonReviewer2 · 2017-11-26
**Nicely written, lack of comparison to other methods, poor results**

**Rating:** 4
**Confidence:** 4

**Review:**

This paper introduce a times-series prediction model that works in two phases. First learns a deterministic mapping from x to y. And then train another net to predict future frames given the input and residual error from the first network. And does sampling for novel inputs by sampling the residual error collected from the training set.

Pros:
The paper is well written and easy to follow.
Good cover of relevant work in sec 3.

Cons
The paper emphasis on the fact the their modeling multi-modal time series distributions, which is almost the case for most of the video sequence data. But unfortunately doesn’t show any results even qualitative like generated samples for other  work on next frame video prediction. The shown samples from model looks extremely, low quality and really hard to see the authors interpretations of it.

There are many baselines missing. One simple one would be what if they only used the f and draw z samples for N(0,1)? VAE is very power latent variable model which also not being compared against. It is not clear what implantation of GAN they are using?.Vanilla GAN is know to be hard to train and there has been many variants recently that overcome some of those difficulties and its mode collapse problem.

---

### Official Review · AnonReviewer3 · 2017-11-28
**interesting model, baselines are weak, not enough signal on it's generalization ability**

**Rating:** 5
**Confidence:** 3

**Review:**

The paper proposes a model for prediction under uncertainty where the separate out deterministic component prediction and uncertain component prediction.
They propose to have a predictor for deterministic information generation using a standard transformer trained via MSE.
For the non-deterministic information, they have a residual predictor that uses a low-dimensional latent space. This low-dim latent space is first predicted from the residual of the (deterministic prediction - groundtruth), and then the low-dim encoding goes into a network that predicts a corrected image.
The subtleness of this work over most other video prediction work is that it isn't conditioned on a labeled latent space (like text to video prediction, for example). Hence inferring a structured latent space is a challenge.
The training procedure follows an alternative minimization in EM style.

The biggest weakness of the paper (and the reason for my final decision) is that the paper completely goes easy on baseline models. It's only baseline is a GAN model that isn't even very convincing (GANs are finicky to train, so is this a badly tuned GAN model? or did you spend a lot of time tuning it?).

Because of the plethora of VAE models used in video prediction [1] (albeit, used with pre-structured latent spaces), there has to be atleast one VAE baseline. Just because such a baseline wasn't previously proposed in literature (in the narrow scope of this problem) doesn't mean it's not an obvious baseline to try. In fact, a VAE would be nicely suited when proposing to work with low-dimensional latent spaces.

The main signal I lack from reading the paper is whether the proposed model actually does better than a reasonable baseline.
If the baselines are stronger and this point is more convincing, I am happy to raise my rating of the paper.

[1] http://openaccess.thecvf.com/content_ICCV_2017/papers/Marwah_Attentive_Semantic_Video_ICCV_2017_paper.pdf

---

### Official Review · AnonReviewer1 · 2017-12-05
**Interesting direction, but alternatives not fully explored and not sure what I learn from experiments.**

**Rating:** 5
**Confidence:** 2

**Review:**

Summary:

I like the general idea of learning "output stochastic" noise models in the paper, but the idea is not fully explored (in terms of reasonable variations and their comparative performance).  I don't fully understand the rationale for the experiments: I cannot speak to the reasons for the GAN's failure (GANs are not easy to train and this seems to be reflected in the results); the newly proposed model seems to improve with samples simply because the evaluation seems to reward the best sample.  I.e., with enough throws, I can always hit the bullseye with a dart even when blindfolded.

Comments:

The model proposes to learn a conditional stochastic deep model by training an output noise model on the input x_i and the residual y_i - g(x_i).  The trained residual function can be used to predict a residual z_i for x_i.  Then for out-of-sample prediction for x*, the paper appears to propose sampling a z uniformly from the training data {z_i}_i (it is not clear from the description on page 3 that this uniformly sampled z* = z_i depends on the actual x* -- as far as I can tell it does not).  The paper does suggest learning a p(z|x) but does not provide implementation details nor experiment with this approach.

I like the idea of learning an "output stochastic" model -- it is much simpler to train than an "input stochastic" model that is more standard in the literature (VAE, GAN) and there are many cases where I think it could be quite reasonable.  However, I don't think the authors explore the idea well enough -- they simply appear to propose a non-parametric way of learning the stochastic model (sampling from the training data z_i's) and do not compare to reasonable alternative approaches.  To start, why not plot the empirical histogram of p(z|x) (for some fixed x's) to get a sense of how well-behaved it is as a distribution.  Second, why not simply propose learning exponential family models where the parameters of these models are (deep nets) conditioned on the input?  One could even start with a simple Gaussian and linear parameterization of the mean and variance in terms of x.  If the contribution of the paper is the "output stochastic" noise model, I think it is worth experimenting with the design options one has with such a model.

The experiments range over 4 video datasets.  PSNR is evaluated on predicted frames -- PSNR does not appear to be explicitly defined but I am taking it to be the metric defined in the 2nd paragraph from the bottom on page 7.  The new model "EEN" is compared to a deterministic model and conditional GAN.  The GAN never seems to perform well -- the authors claim mode collapse, but I wonder if the GAN was simply hard to train in the first place and this is the key reason?  Unsurprisingly (since the EEN noise does not seem to be conditioned on the input), the baseline deterministic model performs quite well.  If I understand what is being evaluated correctly (i.e., best random guess) then I am not surprised the EEN can perform better with enough random samples.  Have we learned anything?

---

### Public Comment · (anonymous) · 2017-11-01
**Question about approach**

How can you be sure that no information of the target seeps through the latent variable $z$? After all, f only needs to learn to undo $\phi$ and that $\phi^{-1}(z) = x - y \Leftrightarrow y = x - \phi^{-1}(z)$.

---

> ### Author Response · Authors · 2017-11-02
> **Re: Question about approach**
>
> Thank you for the question. We actually do want *some* information from the target to seep through the latent variable z, i.e. we would like z to encode information about y which is not predictable from x. For example, if x and y are consecutive images and a new object which was not present in x appears in y from outside the frame (and cannot be predicted from x), we would like this information to be encoded in the latent variable z.
>
> However, it is true that we do not want z to encode information about y which could be predicted from x. The fact that z is of much lower dimension than y forces the network to compress the inherently unpredictable part of y in such a way that z must be combined with x to reconstruct y. This low dimensionality of z prevents the network from learning \phi^{-1}(z) = g(x) - y. In most of our experiments, y is a set of 4 images of dimensions ranging in size from 84x84 to 240x240 (i.e. high-dimensional) whereas z has between 2 and 32 dimensions. In our video generations we condition on a set of frames from the test set, but use z vectors that are extracted from the disjoint training set. If a z vector encoded a lot of information about the specific target used to compute it (rather than general features such as "the paddle moves left" or "a new pipe appears at this height"), then there would be a mismatch between the conditioning frames and the generated frames (for example, different backgrounds), which does not appear to be the case.
>
> We will clarify this in the updated paper.

---

### Public Comment · ~Xin_Yang1 · 2017-11-10
**Evaluation Setting**

While the authors provide some demonstration on Youtube, I'm unsure that the approach really improve the performance,
as there is speculation about evaluation setting.
Although there are lot of works for this area, the authors compares the method with GAN only.
PNSR, the criterion author showed, is not common.

(This is minor comment)
You need not to show all the URLs for each paper in the reference section.

---

> ### Author Response · Authors · 2017-11-11
> **Re: Evaluation Setting**
>
> Thank you for the comment. Although there are indeed many works on video prediction, these are generally deterministic and we are not aware of other work on video data that performs multi-modal prediction other than (Goroshin et. al, 2015) and (Vondrick et. al 2015) mentioned in related work, who perform alternating minimization over latent variables. The focus of these works is different from ours in that Vondrick et. al perform prediction of high-level representations using a pre-trained network (rather than pixels) with the goal of predicting future actions or objects appearing in the video, and Goroshin et. al focus primarily on learning linearized representations and only apply the latent variable version of their model to very simple settings. We tried alternating minimization in early experiments on simple tasks, and found that it performed similarly or worse than our method (in terms of loss) while being considerably slower due to the inner optimization loop and also introduced new hyperparameters to tune such as the learning rate and number of iterations in the inner loop. Comparing to GANs seemed appropriate, since they are a widely used method which can in principle perform multi-modal generations (although as noted in the paper, they can suffer from mode collapse especially in the conditional setting).
>
> To our knowledge, PSNR (along with SSIM) is one of the more common metrics for evaluating video generations, and is used in several works, for example:
>
> https://arxiv.org/pdf/1511.05440.pdf
> https://arxiv.org/pdf/1605.08104.pdf
> https://arxiv.org/pdf/1605.07157.pdf
> https://arxiv.org/pdf/1706.08033.pdf
>
> We also computed SSIM, and found that the EEN performance also increases with the number of samples, although the difference is less pronounced than with PSNR or MSE. Note that the SSIM contains terms which compare statistics taken over windows of the image, meaning that small changes in object location between two images (for example, the paddle moving in Breakout) may not be reflected as much with this metric. However, we can include this metric along with MSE or others if the reviewers think it is appropriate.

---

### Author Response · Authors · 2018-01-05
**Response to Reviews**

We would like to thank the reviewers for the thoughtful reviews and suggestions, they will help make the paper stronger. We have made several changes to the paper which we hope address the reviewers’ concerns.

*We have added three baselines:
-The model proposed by reviewer 2 where we keep f and sample z in N(0, 1), which we call CNN + noise in the paper.
-A conditional VAE model, where the z distribution is a function of phi(x, y).
-A conditional autoencoder model, which is like a VAE but the z’s are deterministic and rather than sampling from N(0, 1) at test time, we use the same non-parametric sampling approach as for the EEN.
We found that the CNN+noise and VAE models experienced mode collapse, which has been observed before in the literature when doing conditional generation when the conditioning variable x contains a lot of signal: the model can achieve large improvements in the prediction loss by learning a deterministic function of x, letting the z distribution go to N(0, 1) to lower the KL term in the loss, and having the rest of the network ignore z. The conditional autoencoder and EEN do not experience this since they do not have a KL term and instead rely on the non-parametric sampling procedure which does not place any assumptions on the latent distribution. We find that the EEN produces generations which are either similar or better (in terms of our performance metric and visual inspection) than the autoencoder.

*We replaced the L2 loss by the L1 loss on the Poke dataset, which we found to improve the generation quality. We also used L1/L2 in the quantitative evaluation, rather than PSNR, so we could evaluate the models on the Poke dataset using the same loss they were trained on.

*Concerning the evaluation metric, we would like to clarify what it does and does not measure (we have updated the description in the paper). It measures whether there exists a sample in the set of generated samples which is close to the test sample in L1/L2 norm. Assuming the conditional distribution P(y|x) is multimodal, the metric does reflect whether the model can generate outputs which cover several modes.  However, it does not say whether all the generated samples are of good quality. For example, if model 1 generates several good samples and then a slightly less good one, and model 2 generates several good samples and then a terrible one, this would not be reflected strongly in this metric. Another way of seeing it is the following: say P(y|x) has modes M1, M2 and the model generates samples in M1, M2 and a third mode M3. This would still get a fairly good score by our metric (although the curves would likely improve a bit more slowly with more samples). However if the model only generates samples in M1, it would get a bad score. We would also like to note that although the evaluation does reward the best guess, all models are given the same number of guesses, therefore we believe it is a fair way to compare models. Models which experience mode collapse will always make the same guess, which leads to poor performance - even being given a huge number of guesses will not improve performance.

*In the previous version of the paper, the results for Flappy Bird and TORCS were for 1 predicted frame, rather than 4 predicted frames. We only include results for 4 predicted frames in the updated paper, as the effects of multi-modality are more pronounced when we look further into the future.

*We found that by increasing the size of the deterministic model we were able to get much better performance on Seaquest (both in terms of loss and generated images). This indicates that there is actually less uncertainty to model in this dataset than previously thought (for example, the agent may be following a policy which is nearly deterministic) , so we removed it to save space.

* We changed the formatting for the Flappy Bird Generations, which we hope makes them more readable.

---

> ### Comment · AnonReviewer1 · 2018-01-10
> **Response**
>
> Overall I am still somewhat confused by the evaluation metric and whether it is testing variance or actually testing mode coverage as the authors claim.  However, this could be due to my misunderstanding and not the fault of the authors.
>
> The paper has improved from the initial submission; I will raise my score to reflect this.  I will also lower my review confidence since I do not feel I have the expertise to judge all of the (new) experimental details of the paper; I defer to other reviewers for their opinion on how much the new experiments bring the paper closer to acceptance.

---

> ### Comment · AnonReviewer2 · 2018-01-11
> **Thanks for the revision, but still weak results.**
>
> Thank you for the revised version of the paper and adding new experiments. Even though the core idea is interesting I am not still convinced of the architecture and the experiment results. One reason could be the fact that video generation might have been wrong choice of task. As there has been many recent works in video sequence generation with recurrent structure that with much better quality than those shown in this work (just one recent example e.g. https://arxiv.org/pdf/1705.10915.pdf). The propose model could be applied to recurrent, and then one can evaluate the true effectiveness of the results here.
> As for the metric, I am not convinced how this could reflect the quality of multi-modality? It could be that all samples are from one mode, and just one of them happens to be of a higher quality than others?
> I increased my score for for better quality of the revisioned paper but I still find the results and arguments weak.

---

### Decision · Program_Chairs · 2018-01-29
**ICLR 2018 Conference Acceptance Decision**

**Decision:**

Reject

**Comment:**

The paper proposes a novel predictive model (e.g., from videos), called error encoding networks, by first learning a deterministic prediction model and then learning to minimize the residual error using latent variables. The latent variables given the sample are estimated by sampling from the prior then updating via gradient descent. The proposed method shows improved performance over the baselines. However, the qualitative results are not fully convincing, possibly because of (1) the limitation of the architecture, (2) suboptimal implementation/tuning of baselines (such as GAN and cVAE).